# communications
# engineering

## COMMENT

# 3D printing has untapped potential for climate mitigation in the cement sector

Ankita Gangotra[1,2], Emanuela Del Gado [1✉] & Joanna I. Lewis [2✉]

Cement-based construction 3D printing (C3DP) has the potential to be a climate solution by promoting cement decarbonization. Here we propose five policy actions that can guide C3DP toward becoming an emission abating tool.

### Cement's emissions problem

Cementitious products, such as concrete and cement, are some of the most used resources in the world, second only to water in terms of consumption. In the coming decades the demand for cementitious materials and new construction will continue to increase due to a rising global population and urbanization. The production of cement currently contributes to approximately 8% of global $CO_2$ emissions. Cement production and use in construction are responsible for up to 77% and 8% of the emissions respectively from the entire lifecycle of cement and of concrete, from production to end-of-life[1]. In cement production, over 50% of the total $CO_2$ emissions are caused by chemical processes due to calcination i.e., the breakdown of limestone to form clinker, the main reactive component in cement products, and the thermal energy used for the process of calcination accounts for 40% of the $CO_2$ emissions. Due to both these inherent material manufacturing processes and to rising global demand, cement is widely considered to be a 'hard-to-abate' sector in a global climate mitigation.

Low-carbon technologies for cement can be implemented through both production-centric policies targeting the cement production process and consumption-centric policies targeting the end-use of cement in construction. Yet all currently proposed solutions fall short of the systemic changes that could be achieved through the scaling of 3D printing applications in the construction sector. For example, in its latest technology roadmap for cement, the International Energy Agency proposes four technological strategies to lower emissions from the production of cement: (1) clinker substitution, (2) energy efficiency, (3) switching to alternative fuels, and (4) emerging carbon capture, utilization, and storage (CCUS) technologies[2]; the same strategies emphasized in other decarbonization strategies for the cement sector[1,3]. All of these roadmaps for decarbonizing the cement sector recognize that while existing low-carbon cement technologies will help with partial $CO_2$ mitigation, newer, breakthrough low-carbon technologies are required to achieve carbon neutrality in the cement and concrete supply chain by mid-century. Cement-based construction 3D printing (C3DP), also known as cement-based additive manufacturing or digital construction, is an emerging technology that, while omitted from all current roadmaps for decarbonizing the cement sector, has a potential of becoming a climate solution for the cement sector, reducing the embodied carbon in new constructions. C3DP is sometimes used to refer to construction 3D printing as a whole, which also includes the use of plastics and metals for construction. In this comment C3DP only refers to cement-based 3D printing.

[1] Department of Physics, Institute for Soft Matter Synthesis and Metrology, Georgetown University, Washington, DC 20057, USA. [2] Science, Technology and International Affairs Program, Edmund A. Walsh School of Foreign Service, Georgetown University, Washington, DC 20057, USA. ✉email: ed610@georgetown.edu; joanna.lewis@georgetown.edu

## Experience to date

The promise of C3DP for use in construction has been gaining traction the last few years, mainly because of its potential to reduce manufacturing materials, times, and costs of construction[4,5]. The technology's ability to create complex structural geometries, reduce labor needs, and be deployed for in-situ construction in harsh environments such as disaster zones and even outer space, presents the promise of a viable and per-haps superior alternative to many types of conventional con-structions. The most common technique for C3DP in commercial applications is the extrusion method in which a gantry system or a robotic arm is used to deposit the 3D printable cementitious mixture layer by layer[4]. While C3DP is still a nascent technology with technical challenges pertaining to the control of flow prop-erties of the printable cementitious mixtures, simultaneous printing of steel reinforcements, and the construction of high-rise buildings, ongoing research is tackling these issues[6]. One area of focus is the development of 3D printed reinforcements (such as steel rebar) for load-bearing components[6,7]. The construction of structural components such as load-bearing pillars has already been demonstrated with digital fabrication which, for example, can use robotic arms to place a steel mesh and then pour the concrete[8]. The use and advantages of C3DP have been demon-strated in a wide range of applications such as in the construction of low-rise residential and commercial buildings, pedestrian and bicycle bridges, military bunkers, and disaster relief shelters, and the construction and repair of roadways and bridges[5]. With over 80 companies offering C3DP solutions worldwide, the concrete 3D printing market was valued at $311 million USD in 2019 and is expected to rise to $41 billion USD by 2027[9].

## An untapped climate solution

Beyond its existing commercial applications, C3DP has the potential to limit greenhouse gas (GHG) emissions from the cement supply chain both in the production of the printable cementitious mixtures and their end-use in new constructions. New low-carbon 3D prin-table cementitious mixtures can be developed with supplementary cementitious materials (SCMs) including industrial wastes (fly ash and slag) and natural materials (clay), alternative binders (geopoly-mers), and recycled materials (conventional concrete)[10], reducing emissions on the production side of the cement supply chain. Through structural design optimization and functional hybridization in construction, C3DP can be used to fabricate structures that use less cementitious materials, decrease the need for formwork, and reduce waste, overall reducing emissions compared to conventional concrete construction[6]. The supply chain can be further dec-arbonized by producing 3D printable cementitious materials using aggregates that are locally available or produced at the construction site[11]. In a life cycle assessment (LCA) study examining emissions from the production to construction stage of a $1\,m^2$ load bearing wall, C3DP was recently found have lower GHG emissions when compared to conventional construction in all scenarios apart from when additional reinforcement was manually installed[12]. Similar LCA comparisons to conventional construction suggest a net reduction in emissions for the C3DP construction of structural pillars[13], bathrooms[14], and residential houses[15]. Multiple studies have also demonstrated that the emissions reduction potential of C3DP is maximized when used to construct complex and unique structures[8,16]. In addition to climate mitigation, C3DP has shown promise in reducing other adverse environmental effects such as eutrophication, acidification, smog formation, and fossil fuel depletion[12]. While more research is needed in the implications of C3DP in the end-of-life phase of a printed structure, this technology also has the potential to increase demolished materials' recyclability and when using locally sourced[17].

Recognizing the potential of C3DP for revolutionizing manu-facturing, governments around the world have started prioritizing this technology in their national advanced manufacturing strategies. In January 2021, the United States Department of Defense released its strategy for the technological advancement of additive manufacturing, including the use of 3D printing in concrete construction, emphasizing the need for further policy development[18]. Also in 2021, Dubai announced a goal to deploy 3D printing in 25% of all new building construction by 2030[19]. China has also included additive manufacturing in its national development strategies for "manufacturing core competitiveness" as outlined in the 14th Five-Year Plan, which has resulted in an increase in government research funding for C3DP and the use of 3D printing in several large-scale construction projects[20]. Yet none of these government plans recognizes the climate mitigation potential of such strategies.

Unlike other forms of 3D printing, emissions associated with the C3DP printing process itself are comparatively low (1–12% of total emissions as shown in refs. [13,21]), so the biggest climate obstacle for this technology is the emissions associated with the production of 3D printable cementitious mixtures[13]. This is similar to conventional construction in which the emissions from material production also outweigh the emissions from energy use in the construction process (which is ~10% of total emissions)[3]. This new C3DP technology therefore presents a unique oppor-tunity to shift the construction industry towards lower-carbon cement mixtures. C3DP requires close control of both (1) the flow properties of the paste during printing and (2) the setting and consolidation of the printed layers beyond what is done in tra-ditional construction. This can only be addressed by designing well-tailored mixtures, making low-carbon cement mixtures that use waste industrial products or calcined clays and other soil-based additives particularly ideal candidates. However, it also would be possible to develop mixtures that are far less climate friendly. While are limited LCAs of printable cementitious mix-tures reported in peer-reviewed literature[22], some studies show that printable cementitious mixtures can contain larger amounts of Portland cement compared to conventional concrete mixtures due to strength and durability constraints, leading to a higher carbon footprint of the materials, others demonstrate the potential of SCMs to lower environmental impacts. For example, studies have found that ternary blends of printable concrete with high levels of supplementary cementitious materials and geopo-lymers have lower carbon footprints (230–295 kg $CO_2/m^3$) compared to printable concrete with large amounts of Ordinary Portland Cement (330–680 kg $CO_2/m^3$)[10], the carbon footprint of 3D printable cementitious mixtures (317–631 kg $CO_2/m^3$) decreases with increase in SCM substitution[23], and including large aggregates in the printable cementitious mixture can lead to a 40–78% reduction in emissions (113–305 kg $CO_2/m^3$) com-pared to other 3D printable mixtures[24]. While imperfections in the interfacial layers in C3DP can affect durability and in turn the sustainability benefits of the technology, the addition of SCMs to printable concrete lowers the carbon footprint while also enhancing the strength and durability of the material[10]. There is therefore a timely opportunity to focus current technology development specifically on low-carbon printable cementitious materials, if policies promoting C3DP incorporate GHG emis-sions criteria from the outset.

## Aligning manufacturing and climate strategies

Without efforts in directing innovations for C3DP towards a low-carbon pathway, the opportunity this technology presents for the climate risks being overlooked. We therefore develop a list of additive manufacturing strategies targeting the production of 3D

**Table 1 Strategy comparison for 3D printable cementitious mixtures across the 5 policy areas.**

| Policy area | Advanced C3DP manufacturing strategy | Climate-optimized C3DP strategy |
|---|---|---|
| 1. Research & Development | Continue funding studies related to material compositions and printing protocols that improve the strength, durability, and long-term performance of 3D printed structures. | • Increase funding for the development of low-carbon 3D printable cementitious mixtures.<br>• Fund studies that improve knowledge of structural design optimization to reduce material use and waste with C3DP. |
| 2. Information Dissemination & Workforce Development | Establish open-source data repositories for industry and academia to share data on reliable materials manufacturing and printing protocols.<br>Commission demonstration projects with long-term performance testing plans to increase trust and the uptake of C3DP in construction projects. | • Encourage reporting and sharing of data on LCA studies, GHG inventories, and other environmental metrics within the open-source repositories.<br>• Conduct environmental assessments and LCA studies in the demonstration projects to test and showcase the environmental impact of C3DP compared to conventional construction.<br>• Encourage construction firms to carry out GHG inventories and share data in open-source repositories. |
| | Establish training programs to produce skilled workers with knowledge of C3DP materials and processes. | • Upskill the workforce by integrating courses on sustainability and the environmental impacts of C3DP into training programs. |
| 3. Standards & Codes | Develop manufacturing standards to ensure the strength and durability of 3D printable cementitious materials. | • Include guidelines in standards for low-carbon 3D printable cementitious mixtures containing materials like slag, fly ash, clay, and geopolymers.<br>• Consider designing emissions product standards for the materials. |
| | Update buildings codes to include safe, replicable, and low-cost 3D printing protocols. | • Integrate design optimization into building codes to encourage an increase in material use efficiency and reduce waste.<br>• Include GHG inventories for printable cementitious materials and printing processes in building codes. |
| 4. Public Procurement & Partnerships | Commission construction projects employing C3DP in publicly funded construction to increase demand. | • Dive innovation by prioritizing the use of low-carbon 3D printable cementitious mixtures and design optimization in the publicly funded C3DP projects. |
| | Establish public-private partnerships with construction firms to increase uptake of C3DP construction. | • Encourage construction firms to use low-carbon 3D printable cementitious mixtures and design optimization in their projects. |
| 5. Financial & Structural Incentives | Offer loans and rebates to conventional cement and concrete producers manufacturing 3D printable cementitious mixtures to increase materials supply and reduce costs. | • Offer additional grants to producers of low-carbon printable cementitious mixtures. |
| | Offer financial and structural incentives to construction firms to increase the uptake of C3DP in construction projects. | • Award tax credits, rebates, and building permits to C3DP construction projects with sustainability objectives such as reduced material use, formwork, and waste. |

Overview of the manufacturing strategies that target the production of 3D printable cementitious mixtures. For each of the 5 policy areas identified, we recommend strategies for using C3DP in construction to bolster climate mitigation and to incorporate greenhouse gas emissions criteria in the technology from the outset.

printable cementitious mixtures and the use of C3DP in construction[25], then recommend detailed policy opportunities that would optimize their climate mitigation potential. First, research and development funding related to material compositions and printing protocols that improve the strength, durability, and long-term performance of 3D printed structures should specifically target the development of low-carbon 3D printable cementitious mixtures as well as the structural design optimization to reduce material use and waste with C3DP. Second, key information should be made publicly available to advance the sharing of crucial information between industry and research communities, with a specific aim of ensuring that demonstration projects with long-term performance testing plans to increase trust and the uptake of C3DP in construction projects are commissioned with requirements for reporting and sharing of data on GHG emissions and other environmental metrics within the open-source repositories. Training to create skilled workers for C3DP must also include sustainability and the environmental impacts of the technology. Third, standards being developed to ensure the strength and durability of 3D printable cementitious materials should include guidelines in standards for low-carbon 3D printable cementitious mixtures containing materials like slag, fly ash, clay, and geopolymers, and consider designing emissions product standards for the materials. In addition, building codes should be updated to include safe, replicable, and low-cost 3D printing protocols that both integrate design optimization into building codes to allow an increase in material use efficiency and reduce waste and include accounting for GHG emissions for printable cementitious materials and printing processes in

building codes. Fourth, government commissioned construction projects using C3DP in publicly funded construction should require the use of low-carbon 3D printable cementitious mixtures in the projects. Public-private partnerships should also be established to increase uptake of C3DP construction. Fifth, loans and rebates should be offered to conventional cement and concrete producers manufacturing 3D printable cementitious mixtures to increase materials supply and reduce costs with a focus on low-carbon printable cementitious mixtures, and financial and structural incentives should be offered to construction firms to increase the use of C3DP in construction projects incorporating sustainability objectives such as reduced material use, formwork, and waste. These manufacturing strategies and associated policy recommendations are detailed in Table 1.

## Conclusion

Cement-based additive manufacturing has the potential of keeping up with the rising demand for new construction while simultaneously becoming a significant climate solution for the cement supply chain—from production to end-use in construction, but only if policy makers are deliberate in their design of policies to advance C3DP manufacturing about also using this technology to abate emissions from the production and consumption of cement. In this paper we recommend optimizing existing and emerging manufacturing strategies with climate policy targets for promoting C3DP.

Research and development funding should continue for investigations of C3DP that improve the mechanical strength,

durability, and long-term performance of 3D printing structures. At the same time, funding needs to be ramped up for the development of new types of sustainable 3D printable cementitious mixtures. Funding is also needed for studies that improve existing knowledge of structural design optimization and hybridization to reduce material use and waste during construction.

For information dissemination, open-sourced repositories should be established which should include sharing of data on life cycle inventories, GHG emissions and other environmental metrics should be encouraged across industry and academia in order to make the environmental impact of C3DP more visible. Different types of large-scale demonstration projects should be commissioned to not only encourage the uptake of the technology, but also showcase the environmental impacts of C3DP materials and printing protocols. Training programs should be established to produce a skilled workforce that not only has knowledge of C3DP materials and processes but is also aware of the sustainability and environmental potential of C3DP.

Material standards need to be developed to set benchmarks for the strength and durability of 3D printable cementitious materials and buildings codes need to include replicable and safe printing protocols for C3DP. The material content or performance standards should also include manufacturing guidelines for low-carbon mixtures with SCMs and geopolymers. Eventually clean product standards accounting for GHG emissions from production could also be developed for these 3D printable materials. Construction building codes need to be updated to allow design optimization and take into account the embodied emissions from the 3D printable mixtures and the energy use from the printing process.

Publicly funded construction projects employing C3DP should be commissioned by governments to increase the demand and awareness of the technology. Such projects should enact public procurement policies prioritizing low-carbon 3D printable cementitious mixtures and design optimization to enhance decarbonization efforts. Similarly, private-public sustainability partnerships could be established to increase the uptake of C3DP and encourage construction firms employing C3DP to carrying out environmental assessments and share data on life cycle inventories and GHG emissions.

Finally, incentives for cement producers to increase the manufacturing of 3D printable cementitious mixtures should include additional loans and grants to encourage the production of low-carbon products. Incentives provided to construction firms for increasing the uptake of C3DP should have provisions for additional awarding rebates, tax credits, and expedited building permits to firms with sustainability objectives for C3DP such as reduced material use and waste.

Acting early to enact such policies could have dramatic implications not only for global carbon emissions but also national competitiveness, the ultimate future of the construction sector, and how we continue to expand our built environment.

## Data availability
All data analyzed during this study are included in this published article and in the references.

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

## Acknowledgements
The authors thank the Georgetown University Earth Commons and the Georgetown University Provost Office for funding.

## Author contributions

E.D.G., J.I.L., and A.G. conceptualized and designed the research. A.G. performed the research and wrote the first draft. E.D.G. and J.I.L. reviewed and edited the paper. J.I.L. and E.D.G. acquired the funding for this work.

## Competing interests

The authors declare no competing interests.
