## [Peer Review File · Communications Engineering]

Reviewers' comments:

Reviewer #1 (Remarks to the Author):

This paper suggests some policies to promote the use 3D concrete printing as a decarbonization tool. The paper is mostly well written. However, its claim about the use of more sustainable mixtures will pave the way for the significant greenhouse gas emissions need to be further supported. Specific comments of these reviewers are:

On page 3, the authors say:

Emissions associated with the 3D printing process itself are comparatively low (10-12% of total emissions), so the biggest climate obstacle for this technology is the emissions associated with the production of 3D printable cementitious mixtures.

This is a claim that can be true but not well supported. It is important that the authors clarify how they come up the contribution of 3D printing to be 10-12% of total.

It is also important if they clarify how this percentage compared with traditional construction.

On page 3, the authors cites a single paper (published in an MDPI journal and not necessarily high quality) to discuss the LCA comparison with 3DCP and conventional construction. If possible, the authors should provide more examples.

The authors says:

"Similar LCA comparisons suggest a net reduction in emissions for the C3DP construction of structural pillars, bathrooms, and residential houses."

But no reference is provided to support this claim.

The total environmental impact of 3DCP is inversely proportional with the service life of structures. We may reduce amount of materials used in the design and also reduce the material impact (i.e., use low carbon cement etc.) but a reduction in service life may negate these benefits. Is there sufficient information on durability of 3DCP?

What about the end of life use of 3DCP? Do you see value more research on this?

Currently, 3DCP is being used mostly for wall-type elements. Will it be used for making other structural elements such as beams?

Question: This paper suggests some policies to promote the use 3D concrete printing as a decarbonization tool. The paper is mostly well written. However, its claim about the use of more sustainable mixtures will pave the way for the significant greenhouse gas emissions need to be further supported.

Answer: This is a great point and we thank the reviewer for the opportunity to provide more information about the support to our claim. While establishing the exact gain with respect to greenhouse gas emissions with more sustainable mixtures is of course subject of ongoing research and analysis, the recent review by Bhattacharjee et al. (Cement and Concrete Composites, vol. 122, 2020) already showed that ternary blends of printable concrete with high levels of supplementary cementitious materials and geopolymers have lower carbon footprints (230-295 kg CO₂/m³) compared to printable concrete with large amounts of Ordinary Portland Cement (330- 680 kg CO₂/m³) and even some conventional ready-mix concrete mixes (350-410 kg CO₂/m³). This result is also supported by the work by Batikha et al. (Automation in Construction, 134, 104087, 2022) which found that the carbon footprint of 3D printable cementitious mixtures (317- 631 kg CO₂/m³) decreases with increase in SCM substitution. In addition, a recent study by da Silva et al. (RILEM International Conference on Concrete and Digital Fabrication, pp. 71-77. Springer, Cham, 2022) also found that including large aggregates in the printable cementitious mixture leads to a 40-78% reduction in emissions (113-305 kg CO₂/m³) compared to other 3D printable mixtures.

We have included the following text on Page 4 of the manuscript:

“For example, studies have found that ternary blends of printable concrete with high levels of supplementary cementitious materials and geopolymers have lower carbon footprints (230-295 kg CO₂/m³) compared to printable concrete with large amounts of Ordinary Portland Cement (330- 680 kg CO₂/m³) [9], the carbon footprint of 3D printable cementitious mixtures (317-631 kg CO₂/m³) decreases with increase in SCM substitution [22], and including large aggregates in the printable cementitious mixture can lead to a 40-78% reduction in emissions (113-305 kg CO₂/m³) compared to other 3D printable mixtures [22].”

We have included the following references to the manuscript:

[23] M. Batikha, R. Jotangia, M. Y. Baaj, and I. Mousleh, ‘3D concrete printing for sustainable and economical construction: A comparative study’, Automation in Construction, vol. 134, p. 104087, Feb. 2022, doi: 10.1016/j.autcon.2021.104087.

[24] W. R. L. da Silva, M. Kaasgaard, and T. J. Andersen, ‘Sustainable 3D Concrete Printing with Large Aggregates’, in Third RILEM International Conference on Concrete and Digital Fabrication, Cham, 2022, pp. 71–77. doi: 10.1007/978-3-031-06116-5_11.

Question: On page 3, the authors say: Emissions associated with the 3D printing process itself are comparatively low (10-12% of total emissions), so the biggest climate obstacle for this technology is the emissions associated with the production of 3D printable cementitious mixtures. This is a claim that can be true but not well supported. It is important that the authors clarify how they come up the

contribution of 3D printing to be 10-12% of total. It is also important if they clarify how this percentage compared with traditional construction.

Answer: The reviewer makes a valuable point, and we appreciate the opportunity to clarify. The emission from the printing process is estimated to be between 1-12% from Life Cycle Assessments studies conducted to account for material and printing process emissions in construction 3D printing as reported in the review by Saade et al. (Journal of Cleaner Production, vol. 244, p. 118803, 2021). This finding is further supported by Muñoz et al. (International Journal of Advanced Manufacturing Technology, pp. 2149–2159, 2021) which found the emissions from the printing process to be ~1%.

We have included the following reference to the manuscript:

[21] M. R. M. Saade, A. Yahia, and B. Amor, 'How has LCA been applied to 3D printing? A systematic literature review and recommendations for future studies', Journal of Cleaner Production, vol. 244, p. 118803, Jan. 2020, doi: 10.1016/j.jclepro.2019.118803.

We note that also for conventional construction the process of construction causes ~10% of the total emissions, while the production of concrete used in building causes over 50% of the emissions as found by Habert et al. (Nature Reviews Earth & Environment, vol. 1, no. 11, pp. 559–573, 2020).

We have included the following text on Page 4 of the manuscript:

"This is similar to conventional construction in which the emissions from material production also outweigh the emissions from energy use in the construction process (which is ~10% of total emissions) [3]."

Question: On page 3, the authors cites a single paper (published in an MDPI journal and not necessarily high quality) to discuss the LCA comparison with 3DCP and conventional construction. If possible, the authors should provide more examples. The authors says: "Similar LCA comparisons suggest a net reduction in emissions for the C3DP construction of structural pillars, bathrooms, and residential houses." But no reference is provided to support this claim.

Answer: Providing further examples is certainly useful and there are a few that we consider particularly relevant. For LCA comparisons of C3DP to conventional construction we have included additional references from Muñoz et al. (International Journal of Advanced Manufacturing Technology, pp. 2149–2159, 2021) which found a 38% reduction in emissions for structural pillars, Weng et al. (Journal of Cleaner Production, vol. 261, p. 121245, 2020) which found a 86% reduction in emissions for bathrooms, and Abdalla et al. (Sustainability, vol. 13, no. 21, p. 11978, Oct. 2021) which found a 53% reduction in emissions for residential homes.

We have included the following references to the manuscript:

[14] Y. Weng et al., 'Comparative economic, environmental and productivity assessment of a concrete bathroom unit fabricated through 3D printing and a precast approach', Journal of Cleaner Production, vol. 261, p. 121245, 2020, doi: 10.1016/j.jclepro.2020.121245. [15] H. Abdalla, K. P. Fattah, M. Abdallah,

and A. K. Tamimi, 'Environmental Footprint and Economics of a Full-Scale 3D-Printed House', *Sustainability*, vol. 13, no. 21, p. 11978, 2021, doi: 10.3390/su132111978.

We have also provided additional examples of LCA studies from Agustí-Juan et al. (*Journal of Cleaner Production*, vol. 154, pp. 330–340, 2017) and Liu et al. (*Chemosphere*, vol. 298, p. 134310, Jul. 2022) which have demonstrated that the emissions reduction achieved with C3DP compared to conventional construction is maximized when used to construct complex structures. We have included the following text on Page 3 of the manuscript: "Multiple studies have also demonstrated that the emissions reduction potential of C3DP is maximized when used to construct complex and unique structures [7], [15]."

We have included the following references to the manuscript:

[8] I. Agustí-Juan, F. Müller, N. Hack, T. Wangler, and G. Habert, 'Potential benefits of digital fabrication for complex structures: Environmental assessment of a robotically fabricated concrete wall', *Journal of Cleaner Production*, vol. 154, pp. 330–340, Jun. 2017, doi: 10.1016/j.jclepro.2017.04.002.

[16] S. Liu, B. Lu, H. Li, Z. Pan, J. Jiang, and S. Qian, 'A comparative study on environmental performance of 3D printing and conventional casting of concrete products with industrial wastes', *Chemosphere*, vol. 298, p. 134310, 2022, doi: 10.1016/j.chemosphere.2022.134310.

We believe that commenting on and referring to these works provides a good range of examples for the LCA comparison between 3DCP and conventional construction. If the reviewer has specific further examples in mind, we would certainly be willing to include them.

Question: The total environmental impact of 3DCP is inversely proportional with the service life of structures. We may reduce amount of materials used in the design and also reduce the material impact (i.e., use low carbon cement etc.) but a reduction in service life may negate these benefits. Is there sufficient information on durability of 3DCP?

Answer: The reviewer makes a very good point. One of the concerns on existing C3DP techniques is precisely that they can detrimentally affect the durability, mainly due to flaws in the interface of the printed layers. That is the reason why fundamental research and further development of these technologies is important and is the subject of ongoing efforts (De Schutter et al. *Cement and Concrete Research*, vol. 112, pp. 25–36, 2018 and Bos et al. *Virtual and Physical Prototyping*, vol. 11, no. 3, pp. 209–225, 2016). Similar efforts, with due differences, are also ongoing in 3D printing technologies for plastics or metals. In all these cases, it has been demonstrated that adjusting the mixtures used and optimizing the printing protocols can lead to much better quality of the interface between the printed layers (Narella et al. *Construction and Building Materials*, 205, 586-601, 2019, Tian et al. *Composites Part A: Applied Science and Manufacturing*, 88, 198- 205, 2016, and Zhang et al. *Journal of Materials Engineering and Performance*, 27(1), 1-13, 2018), and hence to potentially higher durability. As a consequence, it is expected that rapid progresses will be made on the durability of 3D printed materials and structures. For 3DPC, an example is in Bhattacharjee et al. (*Cement and Concrete Composites*, vol.

122, 2020) which has indeed found that the addition of supplementary cementitious materials can enhance the strength and durability of the material while also lowering the carbon footprint.

We have included the following text on Page 4 of the manuscript:

“While imperfections in the interfacial layers in C3DP can affect durability and in turn the sustainability benefits of the technology, the addition of SCMs to printable concrete lowers the carbon footprint while also enhancing the strength and durability of the material [9].”

Question: What about the end of life use of 3DCP? Do you see value more research on this?

Answer: There is indeed some interesting potential. While more research is needed for the implications of C3DP in the end-of-life phase of a structure, the review by Dixit (IOP Conference Series: Earth and Environmental Science, vol. 290, 2019) has found that C3DP has potential sustainability benefits in terms of the reuse and recyclability of demolished materials used in C3DP. The benefits could stem from the use of locally sourced materials and recycling easier.

We have included the following text on Page 3 of the manuscript:

“While more research is needed in the implications of C3DP in the end-of-life phase of a printed structure, this technology also has the potential to increase demolished materials’ recyclability and when using locally sourced [16].”

We have included the following reference to the manuscript:

[17] M. K. Dixit, ‘3-D Printing in Building Construction: A Literature Review of Opportunities and Challenges of Reducing Life Cycle Energy and Carbon of Buildings’, IOP Conference Series: Earth and Environmental Science, vol. 290, no. 1, 2019, doi: 10.1088/1755-1315/290/1/012012.

Question: Currently, 3DCP is being used mostly for wall-type elements. Will it be used for making other structural elements such as beams?

Answer: An area of currently ongoing research is the development of 3D printed reinforcements such as rebar for load bearing, structural component in C3DP. For example, Gebhard et al. (Engineering Structures, vol. 240, p. 112380, 2021) in their recent work have developed a model for designing reinforcement strategies for 3D printed concrete beams. Further, the construction of structural components such as load-bearing pillars has also already been demonstrated with digital fabrication which, for example, can use robotic arms to place a steel mesh and then pour the concrete as shown in Agustí-Juan et al. (Journal of Cleaner Production, vol. 154, pp. 330–340, 2017).

We have included the following text on Pages 2 and 3 of the manuscript:

“One area of focus is the development of 3D printed reinforcements (such as steel rebar) for loadbearing components [6][7]. The construction of structural components such as load-bearing pillars has already been demonstrated with digital fabrication which, for example, can use robotic arms to place a steel mesh and then pour the concrete [8].”

We have included the following references to the manuscript:

[7] L. Gebhard, J. Mata-Falcón, A. Anton, B. Dillenburger, and W. Kaufmann, 'Structural behaviour of 3D printed concrete beams with various reinforcement strategies', *Engineering Structures*, vol. 240, p. 112380, Aug. 2021, doi: 10.1016/j.engstruct.2021.112380.

[8] I. Agustí-Juan, F. Müller, N. Hack, T. Wangler, and G. Habert, 'Potential benefits of digital fabrication for complex structures: Environmental assessment of a robotically fabricated concrete wall', *Journal of Cleaner Production*, vol. 154, pp. 330–340, Jun. 2017, doi: 10.1016/j.jclepro.2017.04.002.

Question: Additionally, from an editorial perspective we believe the manuscript may benefit with a slight change in organization, particularly the “conclusions” section of the manuscript. We recommend breaking the 5 suggested strategies into different paragraphs and expanding on them. Can you please address these changes in your rebuttal file and in the abstract?

Answer: We thank the reviewer for the suggestion. We have expanded on the list of the policy recommendations from Table 1 in the conclusion as requested. We have also separated the 5 strategies into different paragraphs and addressed these changes in the abstract, in addition to the changes made to the main body of the paper.

All revisions to the manuscript are highlighted in blue in a file provided separately with this resubmission. Apart for the revisions mentioned in this document, the manuscript also includes some, minor typographical revisions.

REVIEWERS' COMMENTS:

Reviewer #1 (Remarks to the Author):

The authors has satisfactorily responded to this reviewer's comments. The reviewer does not have additional comments.